



# Dynamics of hydrological model parameters: calibration and reliability

Tian Lan[1], Kairong Lin[1,2,3], Xuezhi Tan[1,2,3], Chong−Yu Xu[4], Xiaohong Chen[1,2,3]

[1]Center for Water Resources and Environment, Sun Yat-sen University, Guangzhou, 510275, China.
[2]Guangdong Engineering Technology Research Center of Water Security Regulation and Control for Southern China, Guangzhou 510275, China.
[3]School of Civil Engineering, Sun Yat-sen University, Guangzhou, 510275, China.
[4]Department of Geosciences, University of Oslo, P.O. Box 1047, Blindern, 0316 Oslo, Norway

*Correspondence to*: Kairong Lin (linkr@mail.sysu.edu.cn)

**Abstract.** It has been demonstrated that the dynamics of hydrological model parameters based on dynamic catchment behavior significantly improves the accuracy and robustness of conventional models. However, the calibration for the dynamization of parameter set involves critical components of hydrological models, including parameters, objective functions, state variables, and fluxes, which usually are ignored. Hence, it is essential to design a reliable calibration scheme regarding these components. In this study, we compared and evaluate five calibration schemes with respect to multi-metric evaluation, dynamized parameter
values, fluxes, and state variables. Furthermore, a simple and effective tool was designed to assess the reliability of the dynamized parameter set. The tool evaluates the convergence processes for global optimization algorithms using violin plots (ECP-VP), effectively describes the convergence behaviour in individual parameter spaces. The different types of violin plots can well match to all possible properties of fitness landscapes. The results showed that the reasons for poor model performance included time-invariant parameters oversimplifying the dynamic response modes of the model, the high-dimensionality
disaster of parameters, the abrupt shifts of the parameter set, and the complicated correlations among parameters. The proposed calibration scheme overcome these issues, characterized the dynamic behaviour of catchments, and improved the model performance. Additionally, the designed ECP-VP tool effectively assessed the reliability of the dynamic parameter set, providing an indication on recognizing the dominant response modes of hydrological models in different sub-periods or catchments with the distinguishing catchment characteristics.

## 1 Introduction

Hydrological modelling is an essential tool for understanding the hydrological processes of a catchment and for forecasting streamflow(Liu et al., 2018;Turner et al., 2017;Delorit et al., 2017; Fenicia et al., 2014;Fenicia et al., 2018; Hublart et al., 2016;Liu et al., 2015;Höge et al., 2018;Sarrazin et al., 2016;Wi et al., 2015;Herman et al., 2013;Wagener et al., 2003;Wagener et al., 2001;Madsen, 2000). However, the paucity of progress in model development is (in part) due to structural inadequacy,
which is one reason for poor simulation performance. For example, key catchment dynamics are not adequately represented by models, especially under changing climate and land-surface conditions (Xiong et al., 2019;Deng et al., 2018;Dakhlaoui et





al., 2017;Sarhadi et al., 2016;Pathiraja et al., 2016;Ouyang et al., 2016;Deng et al., 2016). The dynamic components in hydrological models are oversimplified due to a poor understanding of their physical mechanisms. Namely, a recognized issue in hydrological simulations is that a unique parameter set that is obtained by calibrating hydrological models only represents the average hydrological processes. However, the assumption, i.e., the time-invariant parameters, is usually unreasonable due

to they cannot actually represent the dynamic response modes of the catchments processes (Pathiraja et al., 2018;Fowler et al., 2018;Zhao et al., 2017;Kim and Han, 2017;Golmohammadi et al., 2017;Delorit et al., 2017;Chen et al., 2017).

The dynamics of hydrological model parameters may be a type of compensation for models that are missing key processes that occur in catchments with climate- and land surface-related changes (Xiong et al., 2019;Deng et al., 2018;Wang et al., 2017b;Dakhlaoui et al., 2017;Sarhadi et al., 2016;Pathiraja et al., 2016;Ouyang et al., 2016;Deng et al., 2016;Todorovic and

Plavsic, 2015). However, a critical but often overlooked issue related to dynamic parameters is that there are intricately linear or nonlinear correlations among hydrological model parameters (Wagener and Kollat, 2007). It has been conclusively demonstrated that the optimal parameters in hydrological models should not be considered as individual parameters but instead as parameter vector "teams" (i.e., overall parameter set). It is evident that the dynamics of the individual parameters may not represent the time-varying properties of river catchments due to the "compensation" (i.e., correlations) among parameters.

Therefore, it is difficult to determine the temporal changes in the individual parameters over the dynamic catchment features (Höge et al., 2018;R. et al., 2010;Bárdossy and Singh, 2008;Bárdossy, 2007;Huang, 2005).

For those reasons, the most approach for dynamics of hydrological model parameters is that the calibration period is partitioned into different sub-periods based on the temporal dynamic catchment characteristics. The parameter set in each sub-period is respectively optimized to obtain the dynamic parameter set. That is, the sub-period calibration based on the dynamic catchment

behavior is designed to capture the temporal variations of catchment characteristics, compensating the structural inadequacy. Previous studies have demonstrated that sub-period calibration significantly improved the accuracy and robustness of conventional models and the temporally dynamic parameter set exerts a significant impact on hydrological simulations (Lan et al., 2018;Zhao et al., 2017;Kim and Han, 2017;Zhang et al., 2011;De Vos et al., 2010;Gupta et al., 2009;Choi and Beven, 2007;van Griensven et al., 2006;Freer et al., 2003). Zhang et al. (2011) proposed a general multi-period calibration approach

for improving the performance of hydrological models based on the fuzzy c-means clustering technique (FCM) under time-varying climatic conditions. The results indicated that model simulations using parameters obtained from the multi-period calibration approach exhibited considerable improvements over those from the conventional single-period model. Kim and Han (2017) performed a sub-annual calibration with fixed months in each year. The results demonstrated that the seasonal calibration greatly improved simulation accuracy. Lan et al. (2018) applied a clustering pre-processing (CPP) framework for

sub-annual calibration of hydrological models based on climate-land surface variations. The results showed that the sub-annual calibration with the CPP framework exhibited significant improvements in performance.





Even though the sub-period calibration has performed well for the dynamics of hydrological model parameters, a number of fundamental problems still need to be addressed. The sub-period calibration involves the critical components of models, including parameters, objective functions, state variables, and fluxes. The improper handling of those components of models may cause poorer model performance. However, very few studies further investigated internal components when implementing

the sub-period calibration. Thus, objective 1 in this work is to develop an effective sub-period calibration scheme for dynamics of hydrological model parameters.

Another critical issue is whether the parameter set dynamized by sub-period calibration is reliable. The parameters in hydrological models cannot be measured directly and are estimated through calibration using global optimization algorithms. Hence, the convergence performance of global optimization algorithms determines the finally optimal parameter values

(Gomez, 2019;Weise, 2009). Namely, the failure of convergence means that the global optimum cannot be determined. If the optimal parameter values derived from global optimization algorithms are abnormal or unreasonable, the optimal results do not accurately represent the hydrological processes in a catchment. Hence, the analysis and discussion of the dynamic parameter set from the proposed sub-period calibration would be invalid and meaningless. In this regard, we further assess the reliability of the optimized dynamic parameters by evaluating the convergence performance of the global optimization

algorithm for hydrological models.

Several techniques have been developed in previous studies to determine the convergence behaviour of hydrological models. (1) In the response surface methodology, the objective function values (i.e., fitness values) obtained at grid points are used to construct mesh surface plots. However, the objective function surface in the multi-parameter space may be uneven and discontinuous. The derivatives may be discontinuous and vary in an unpredictable manner in the parameter space. Additionally,

the response surface in the multi-parameter space is difficult to be visualized. Due to these problems (including the visualization of high-dimensional parameters, rough response surfaces with discontinuous derivatives, the poor and varying sensitivities of the response surface in the optimum region, and non-convex response surfaces), the response surface, especially in the multi-parameter space of hydrological models, cannot usually be used to assess the convergence performance in many cases (Duan et al., 1994;Duan et al., 1993;Duan et al., 1992). (2) A second method is the evaluation of the objective function

using the start and end points. The $x$-axis represents the number of objective function evaluations and the $y$-axis represents the parameter values. The different start and end parameter values for the optimization can be determined but the detailed convergence processes and convergence speeds cannot be assessed. (3) Derrac et al. (2014) analysed the objective function values at different cut-off points (i.e., different steps of the search) to evaluate the convergence behaviour. This provided a good representation of the convergence speed but does not explain the reason for the convergence success or failure. Taking

into account the limitations of traditional tools for evaluating the convergence behaviour (Duan et al., 1992;Sorooshian et al., 1993;Duan et al., 1994;Cooper et al., 1997;Gupta et al., 1998;Vrugt et al., 2005;Weise, 2009;Zhang et al., 2009;Sun et al., 2012;Arora and Singh, 2013;Derrac et al., 2014;Piotrowski et al., 2017;Gomez, 2019), objective 2 in this work is to design a

**Hydrology and Earth System Sciences**

**Discussions**

EGU

simple and powerful approach to evaluate the convergence performance of global optimization algorithms for hydrological models and to determine the reliability of the dynamic parameter set.

The paper is structured as follows. Section 2 presents the clustering (partition) results of sub-periods from our previous research as the background in this work. Section 3 compares the five calibration schemes for dynamic parameters in hydrological

models and their evaluation system. Moreover, a simple and powerful tool to assess the reliability of the optimized dynamic parameter set is introduced. Section 4 presents the evaluation results of different schemes and the reliability of optimized dynamic parameter set, followed by a discussion for the possible reasons for poor model performance. Section 5 summarizes the key conclusions of the study and the outlines directions for future research.

## 2 Background

Our previous research focused on the reasonable clustering (partition) of sub-periods based on dynamic catchment characteristics. Namely, a CPP (clustering preprocessing) framework was proposed to capture the temporal changes in catchment characteristics. Historical data (1980–1987 for calibration and 1988–1990 for validation) in three basins (including the Hanzhong basin, Mumahe basin, and Xunhe basin, see Figure 1a) are partitioned into different sub-periods for calibration. The clustering results (see Figure 1b) are applied in this study. More detailed descriptions of the clustering refer to Lan et al.

(2018). The elaboration of the case studies is presented in Section 1 of the supporting information.

## 3 Methodology

### 3.1 Sub-period calibration

#### 3.1.1 Sub-period calibration schemes

In order to explore the effect of different components of models (including (1) objective functions, (2) parameters, (3) state

variables and fluxes) on the model performance, five calibration schemes are designed and compared, as follows. Their schematic illustration is shown in Figure 2a.

*Scheme 0.* The parameters do not change during the entire calibration and validation periods.

*Scheme 1.* In the calibration period, the parameter sets in different sub-periods are optimized simultaneously by minimizing one objective function (see Figure 2a). For example, 20 parameters (5 parameters of a hydrological model in 4 sub-periods)

are optimized with one run simultaneously. The transition of state variables and fluxes between two consecutive sub-periods is achieved by considering the ending values of the former period as the initial values of the next period. The parameter set between two consecutive sub-periods is updated accordingly. In the validation period, inputs are covered in a run with the parameter sets in different sub-periods. The transition ways of parameters, state variables, and fluxes between two consecutive sub-periods are same with the transition in the calibration period.



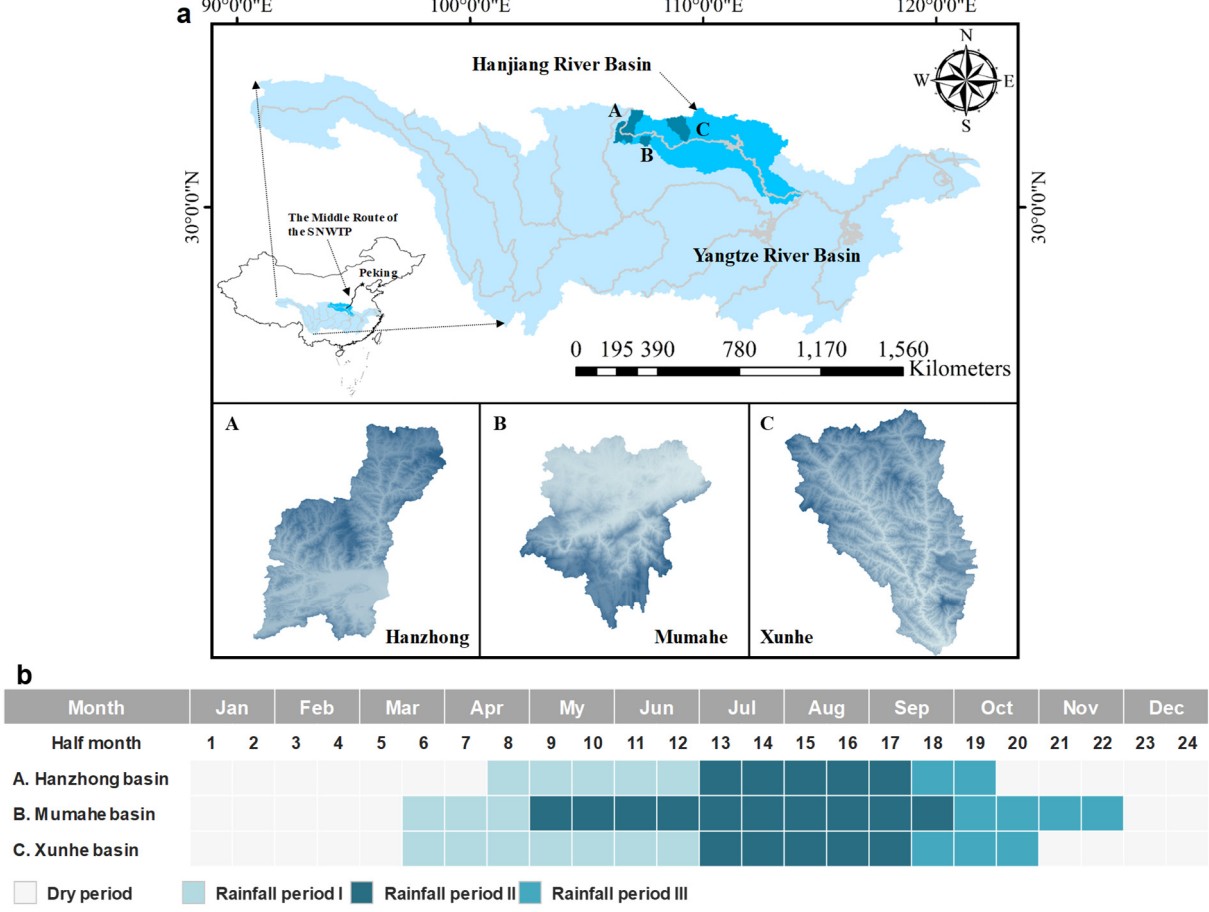

**Figure 1: a. Locations of the study region, the Hanjiang River and its three major tributaries considered in this study, i.e., the Hanzhong, Mumahe and Xunhe Rivers. SNWTP denotes the South-to-North Water Transfer Project in China; b. the heat map of the clustering results.**

5    *Note.* **The sub-periods include the dry period, rainfall period I, rainfall period II, and rainfall period III. In the dry period, both the total amount and the variance values of all the precipitation series reach the minimum. In contrast, the total amount and the variance values of the precipitation series in the rainfall II (wettest period) reach the maximum and the frequency of heavy rain is highest. In the two normal sub-annual periods (rainfall period I and rainfall period III), the climatic patterns are similar but the streamflow volume is higher in rainfall period III than in rainfall period I. The reason is that higher antecedent soil moisture content contributed**
10    **to higher runoff in rainfall period III than in rainfall period I.**

*Scheme 2.* In the calibration period, the specified dynamic parameter and other fixed parameters in different sub-periods are optimized simultaneously by minimizing one objective function (see Figure 2a). For example, 8 parameters (one specified parameter in 4 sub-periods and another four fixed parameters) are optimized with one run simultaneously. The transition ways of parameters, state variables, and fluxes between two consecutive sub-periods are same with the transition in the calibration

15   period of Scheme 1. In the validation period, inputs are covered in a run with the specified dynamic parameter and other fixed parameters in different sub-periods. The transition ways of parameters, state variables, and fluxes between two consecutive sub-periods are same with the transition in the calibration period.





*Scheme 3.* In the calibration period, only the data from individual sub-periods are used for minimizing the objective function, while the model is running for the whole period. For example, five parameters of a hydrological model in 4 sub-periods are optimized with four runs, respectively. The calibrated flow data from each separate sub-period are then combined to compare the observed flow. In the validation period, the transition ways of parameters, state variables, and fluxes between two

consecutive sub-periods are same with the transition in the validation period of Scheme 1 (see Figure 2a).

*Scheme 4.* In the calibration period, the model running is the same as the running in the calibration period of Scheme 3. In the validation period, the simulated flow data from each separate sub-period are then combined to compare the observed flow (see Figure 2a).

For illustration purposes, the HYMOD model (Moore, 1985;Wagener et al., 2001;Vrugt et al., 2002;Yadav et al., 2007;De Vos

et al., 2010;Pathiraja et al., 2018), which is a simple and commonly lumped rainfall-runoff model, is utilized. The details of the model parameters, state variables, and fluxes are presented in Figure 2c and Table 1. More introduction for HYMOD is present in Section 2 of the supporting information. The objective function is defined as the combination of the Nash-Sutcliffe efficiency index (NSE) and the logarithmic transformation (LNSE) (Nash and Sutcliffe, 1970) (i.e., objective function value = $1 - 0.5 \cdot (\text{NSE} + \text{LNSE})$). The NSE is sensitive to the discharge dynamics and the LNSE emphasizes the low flows due to

log of the discharge (Nash and Sutcliffe, 1970;Guntner et al., 1999;Kiptala et al., 2014;Nijzink et al., 2016). In addition, the simulations in the calibration period with a warm period of one year are continuous and the warm period is three months in the validation period to reduce the influence of the initial values on the state variables.

### 3.1.2 Evaluation system

The performance of the five schemes with the same set of the shuffled complex evolution method developed at the University

of Arizona (SCE-UA) (see Section 3 in the supporting information) is evaluated in terms of the multi-metric evaluation, the dynamic parameter values, and the state variables and flux, as illustrated in Figure 2b. (1) A multi-metric evaluation framework (including the NSE, LNSE, and the 5-segment flow duration curve (5FDC) with the root mean squared error (RMSE) (Pfannerstill et al., 2014)) is used to assess the performance of the model runs in the different parts (very low, low, medium, high, and very high flow phases) of the hydrograph in the calibration and validation periods. The detailed descriptions are

listed in Table S1 in the supporting information. Furthermore, the differences in these metrics in the calibration-validation period are calculated to assess the transferability of the optimized parameters (i.e., the robustness of model performance). Here, the transferability of parameters in time is regarded as one of the main requirements for the successful validation of model performance, i.e., the parameters that were calibrated based on the dataset in the calibration period also provide good simulation results in the validation period (Gharari et al., 2013;Klemeš, 1986). (2) The dynamic parameter sets optimized by Scheme 1-4

are investigated to determine the underlying physical mechanisms of the model running based on dynamic catchment characteristics. It is worth noting that the complicated correlations among the parameters make it difficult to identify the changes in individual parameters over time (Bárdossy, 2007). Therefore, we investigated the parameters for different model







Figure 2: a. Schematic illustration of five schemes and their b. evaluation system; c. Hymod structure.





response modes (for example, soil moisture accounting and routing in Figure 2c). (3) The abrupt shifts of state variables and fluxes between two consecutive sub-periods may crash the simulation results of models. Hence, all the state variables and fluxes obtained by different schemes are further investigated, and their underlying physical mechanisms are discussed (Kim and Han, 2017).

**Table 1. Definition of parameters, state variables, and fluxes used in HYMOD model (Wagener et al., 2001).**

| Label | Property | Range | Description |
|---|---|---|---|
| $H_{uz}$ | Parameter | 0-1000 [mm] | Maximum height of soil moisture accounting tank |
| $B$ | Parameter | 0-1.99 | Scaled distribution function shape |
| alpha | Parameter | 0-0.99 | Quick/slow split |
| $K_q$ | Parameter | 0-0.99 | Quick-flow routing tanks' rate |
| $K_s$ | Parameter | 0-0.99 | Slow-flow routing tank's rate |
| $XH_{uz}$ | State variable | [mm] | Upper zone soil moisture tank state height |
| $XC_{uz}$ | State variable | [mm] | Upper zone soil moisture tank state contents |
| $X_q$ | State variable | [mm] | Quick-flow tank states contents |
| $X_s$ | State variable | [mm] | Slow-flow tank state contents |
| $AE$ | Flux | [mm] | Actual evapotranspiration flux |
| $OV$ | Flux | [mm] | Precipitation excess flux |
| $Q_q$ | Flux | [mm] | Quick-flow flux |
| $Q_s$ | Flux | [mm] | Slow-flow flux |
| $Q_{sim}$ | Flux | [mm] | Total streamflow flux |

## 3.2 Reliability evaluation

### 3.2.1 Problems in global optimization

We use a global optimization algorithm, i.e., SCE-UA as an example (Duan et al., 1992;Duan et al., 1994;Duan et al., 1993;Eckhardt and Arnold, 2001;Khakbaz and Kazeminezhad, 2012); its strategy is presented in Figure 3c. There are three convergence criteria in the SCE-UA. (1) The maximum number of function evaluations has been reached. This means that the optimization algorithm does not converge at the end of the run. (2) The population has prematurely converged to a pre-specified small geometric range. This indicates the global convergence has failed. (3) The improvement for the best point in the last loop is less than the specified threshold. This indicates that global convergence has been achieved. Certain convergence criteria of the global optimization algorithm lead to possible failure in finding the global optimum, which may lead to abnormal or unreasonable optimal parameters. Namely, the optimized parameters cannot effectively characterize the physical system of the hydrological models (or catchment response behaviour).

The reasons for the deterioration of the convergence performance in global optimization algorithms include (1) the high-dimensional and nonlinear nature of hydrological models with correlated parameters, (2) dataset errors, which generally increase the number of local optima, (3) the limitation on the maximum number of trials (the maximum number of objective function evaluations is set as 10,000 in this study), (4) the population of points converging into a pre-specified small parameter space (its measure is less than $10^{-3}$% of the feasible space), i.e., premature convergence, and (5) a combination of the aforementioned reasons (Cooper et al., 1997;Duan et al., 1992;Duan et al., 1994;Duan et al., 1993;Gupta et al., 1998;Sorooshian et al., 1993;Vrugt et al., 2005;Zhang et al., 2009).



Fitness landscapes are a very powerful metaphor for visualizing the convergence processes in global optimization (Aldrich, 1997;Dawkins, 1997;Gavrilets, 2004;Kauffman, 1993;Mitchell, 1998;Wright, 1932). Some intuitive sketches of fitness landscapes with possible properties are shown in Figure 3b. Their horizontal axis denotes parameter values; the vertical axis denotes the objective function values. The direction of the arrow represents the direction of evolution. The possible properties

are elaborated; these include (a) Best case: an optimization process is ideal for estimating the globally optimal parameters. (b) Low variation: an optimization process with low variation is fair for estimating the globally optimal parameters. (c) Premature convergence: an optimization process has prematurely converged to a local optimum if it is no longer able to explore other parts of the search space than the area currently being examined and there exists another region that contains a superior solution. (d) Ruggedness: if the objective function values are unsteady or fluctuating, i.e., increasing or decreasing, it is too complicated

for the optimization process to find the right directions to proceed. In short, ruggedness is multi-modality plus steep ascends and descends in the fitness landscape. (e) Deceptiveness: the gradient of the deceptive objective function values leads the optimizer away from the optima. (f) Neutrality: the outcome of the application of a search operation to an element of the search space is neutral if it yields no change in the objective function values. (g) Needle-In-A-Haystack: the optimum occurs as an isolated spike in a plane, representing the occurrences of extreme ruggedness combined with a general lack of information in

the fitness landscape. (h) Nightmare: the optimum is difficult to achieve in an approximate plane. More detailed interpretations for fitness landscapes with possible properties refer to Weise (2009).

### 3.2.2 A tool for reliability evaluation

In order to overcome the limitation of traditional tools for evaluating the convergence behaviour of the global optimization algorithm for hydrological models, a simple and powerful approach is proposed to Evaluate the Convergence Performance

using Violin Plots (ECP-VP). It is designed to represent the possible features of the aforementioned fitness landscapes and visualize the convergence behaviour in multi-parameter space. The strategy (see Figure 3) is as follows:

1.   The end of each evolution loop in the optimization process is regarded as a cut-off point (see as shown in Figure 3a and 3c). In this regard, the parameter set with the best objective function value in each evolution loop is recorded in the "convergence process". That is, the number of elements in the convergence process is the number of evolution loops in

25        an operation process. The final optimum is obtained at the end of the run. It should be noted that as the number of iterations increases, the objective function values are gradually decreased. Namely, the convergence process evolves toward minimizing the objective function values (see Figure 3d).

2.   The violin plots, which are an excellent tool to visualize the kernel density distribution of the data points (Hintze and Nelson, 1998;Piel et al., 2010), are used to configure the convergence process in the individual parameter spaces, i.e., the

30        probability distributions of the violin plots are used to represent the possible properties of the fitness landscapes. The anatomy of the violin plot and the associated information can be found in the supporting information (Section 4). As shown in Figure 3a, the horizontal axis of the violin plot denotes parameter values; the vertical axis denotes the probability





values. The violin plots effectively present the probability distribution of elements of the search space. With the same cut-off points in individual parameter spaces, the thinner distribution type of violin plot indicates that the optimization process at the highest speed finds the global optimum. The bimodal distribution of the violin plot implied that the search is indecisive due to the prominent interfere with local optimum. The search may fail to find a global optimum. The multimodal or flat distribution signifies that the search at the lowest speed in unsteady or fluctuating ways is difficult to find the final optimum. Furthermore, Figure 3a and 3b demonstrate that the convergence performance is positively correlated with convergence speed in the global optimization algorithm.

3. With an adequate parameter space and sufficient density of coverage, the four types of distributions of violin plots (see Figure 3a) in the ECP-VP match the properties' sketches of the fitness landscapes in Figure 3b. The possible convergence performance levels and their candidate mechanisms are interpreted as (I) Unimodal distribution (Figure 3a): an ideal global convergence process is used to estimate the best solution. The unimodal distribution matches two types of fitness landscape sketches including the best case and low variation (Figure 3b). (II) Bimodal distribution (Figure 3a): there are two main local optima and the distance to the two local convergence regions is far. It becomes more complicated for the optimization process to find the global optimum and the premature convergence to a local optimum may occur (Duan et al., 1992;Duan et al., 1993;Duan et al., 1994;Weise, 2009;Sun et al., 2012;Derrac et al., 2014;Gomez, 2019). The bimodal distribution symbolizes the two types of fitness landscapes sketches including the multimodal and deceptive types (Figure 3b). (III) Multimodal distribution (Figure 3a): the response surface may be multi-modal plus steep ascends and descends. This means that multiple local optima exist. With the maze of minor local optima, the calibration algorithm may fail to reach the global optimum. Because the minor optima may be found quite far from the global optimum, the search may terminate prematurely without finding an approximate solution (Dakhlaoui et al., 2017;Duan et al., 1994;Duan et al., 1993;Duan et al., 1992). The multimodal distribution matches the three types of fitness landscapes sketches including the multimodal, rugged, and deceptive types (Figure 3b). (IV) Flat distribution (Figure 3a): this is similar to the multimodal distribution and its surface may be noisy. The very poor sensitivity of the objective function to the parameter fluctuation causes weak convergence of the parameter (Duan et al., 1992;Duan et al., 1993;Duan et al., 1994;Dakhlaoui et al., 2017;Rahnamay Naeini et al., 2018;Vrugt and Beven, 2018). The flat distribution matches the three types of fitness landscapes sketches including the neutral, needle-in-a-haystack, and nightmare types (Figure 3b).

4. Furthermore, the convergence speed could also be assessed using the ECP-VP. With the same number of loops, the objective values are decreasing at multiple steps of the cut-off points (see Figure 3d). It can be concluded that the convergence speeds in Figure 3a (I)-(IV) are decreasing.

5. It needs to be clarified that there are large amounts of objective function evaluations in each loop. For example, the competitive complex evolution (CCE) process in a loop process includes a generating offspring step, a reflection step, a mutation step, and a contraction step. Those steps in the CCE have numerous objective function evaluations (Duan et al., 1994;Duan et al., 1993;Duan et al., 1992) (see Figure 3c). Actually, if all the objective function values of the





corresponding points (i.e., semi-random points) in one optimization process are used, the convergence processes cannot be accurately visualized and analysed. However, the loops represent the multiple recalibrations in one calibration process. The loops do not stop until the convergence criteria are satisfied. To conclude, the result of each loop is suitable for use as a cut-off point.

**Figure 3: a. Evaluation of the convergence processes using violin plots (ECP-VP); b. all possible properties of fitness landscapes; c. SCE-UA algorithm; d. evaluation of convergence speed.**





In summary, the violin plots in the ECP-VP framework are able to well configure the convergence process regarding the possible properties of the fitness landscapes in the global convergence. The ECP-VP, as an easy and effective tool, could be applied to assess the convergence performance and speed in the individual parameter spaces.

## 4 Results and discussion

### 4.1 Evaluation of calibration schemes

#### 4.1.1 Multi-metric evaluation

The evaluation results of five schemes based on the multi-metric evaluation framework in the Hanzhong basin are shown in Figure 4. In Scheme 1, it results in the worst model performance, i.e., all metric values are much higher than 1 in the calibration period and validation period (the lower metric values denote better model performance). In Scheme 2, its model performance is only slightly better than that of Scheme 0, which indicates that the model performance has not improved prominently. In Scheme 3, the overall model performance is higher than that of the other schemes in the calibration period. For example, its NSE and LNSE are 45.3% and 13.8% considerably higher than the metric values of Scheme 0, respectively. The 5FDC metrics also show that Scheme 3 performs best in all (very low, low, medium, high and very high) flow phases in the calibration period. However, the model performance of Scheme 3 in the validation period is only slightly better than that of Scheme 0. In Scheme 4, Scheme 4 is the same as Scheme 3 in the calibration period. Nevertheless, the overall model performance in Scheme 4 is higher than that of the other schemes in the validation period.

Furthermore, the transferability of the optimized parameters in all schemes is analysed. As illustrated in Figure 4, Scheme 4 significantly narrows the differences between the overall metrics in the calibration-validation period. Scheme 3 exhibits the largest differences in the metrics in the calibration-validation period. The transferability performance of the other schemes is between those of Scheme 4 and Scheme 3.

To summarize, Scheme 4 not only significantly improves the overall performance under different flow conditions (high, middle, or low) in the calibration period and validation period but also exhibits good transferability of the model parameters in the calibration-validation period. Scheme 3 exhibits good performance in the calibration period but does not perform well in the validation period, indicating that the temporal transferability of the model in Scheme 3 is poor. Scheme 2 does not improve the model performance in the calibration period or validation period compared to Scheme 0. Scheme 1 results in very poor model performance. Moreover, the evaluation results of the multi-metric framework in Scheme 0 and Scheme 4 in the Mumahe basin and Xunhe basin are listed in Table S2 and Table S4 in the supporting information. The results are similar to those of the Hanzhong basin. The results will be discussed in section 4.2.





**Figure 4: a. Evaluation results of five schemes based on the multi-metric evaluation framework in the Hanzhong basin; b. five segments of the flow duration curve (FDC) are presented to describe different flow characteristics.**

### 4.1.2 Dynamized parameter values

5   Dynamized parameter values optimized by the four sub-period calibration schemes (Scheme 1-4) in the Hanzhong basin are shown in Figure 5. In Scheme 1, the parameters $H_{UZ}$ (maximum height of soil moisture accounting mode) and $B$ (scaled distribution function shape) in the soil moisture accounting mode (see Figure 2c) (Moore, 1985;Vrugt et al., 2002) are quite distinct and no regular patterns are observed in all schemes. The parameters $\alpha$ (quick split), $K_q$ (quick-flow routing tanks' rate) and $K_s$ (slow-flow routing tank's rate) in the slow- and quick-flow routing mode exhibit no prominent changes. In Scheme 2,

10   the dynamic parameter $K_q$ and the other fixed parameters are optimized. However, the responses of dynamized $K_q$ to the dynamic catchment characteristics are ambiguous. In short, there are no parameters with high identifiability in Scheme 1 or Scheme 2. In Scheme 4 (which is the same as Scheme 3 in the calibration period), the parameter $K_q$ exhibits no significantly regular changes, but the $K_s$ accurately describes the discernible model responses in the sub-periods with different catchment characteristics. That is, its value is lowest in the dry period and highest in the rainfall period II (wettest period) but the parameter





is not able to distinguish the two normal periods (i.e., rainfall period I and rainfall period III). The main reason is that, most of the excess streamflow in three rainfall periods is diverted to the slow-flow tank, i.e., the quick-flow tanks do not play no role because the $\alpha$ values are close to the minimum of the range of values.

In sum, Scheme 4 with good model performance has more high-identifiability parameters (including $\alpha$, $K_q$ and $K_s$). The results indicate that Scheme 4 enhances the identification of the dominant parameters and their responses to catchment processes. In turn, the dynamic features of the parameters demonstrate the necessity of sub-period calibration (i.e., invariant parameter set replaced by temporally dynamic parameter set). The dynamic parameter sets optimized by Scheme 0 and Scheme 4 in the Mumahe basin and Xunhe basin are listed in Table S3 and Table S5. The results are similar to those of the Hanzhong basin. The results will be discussed in section 4.2.

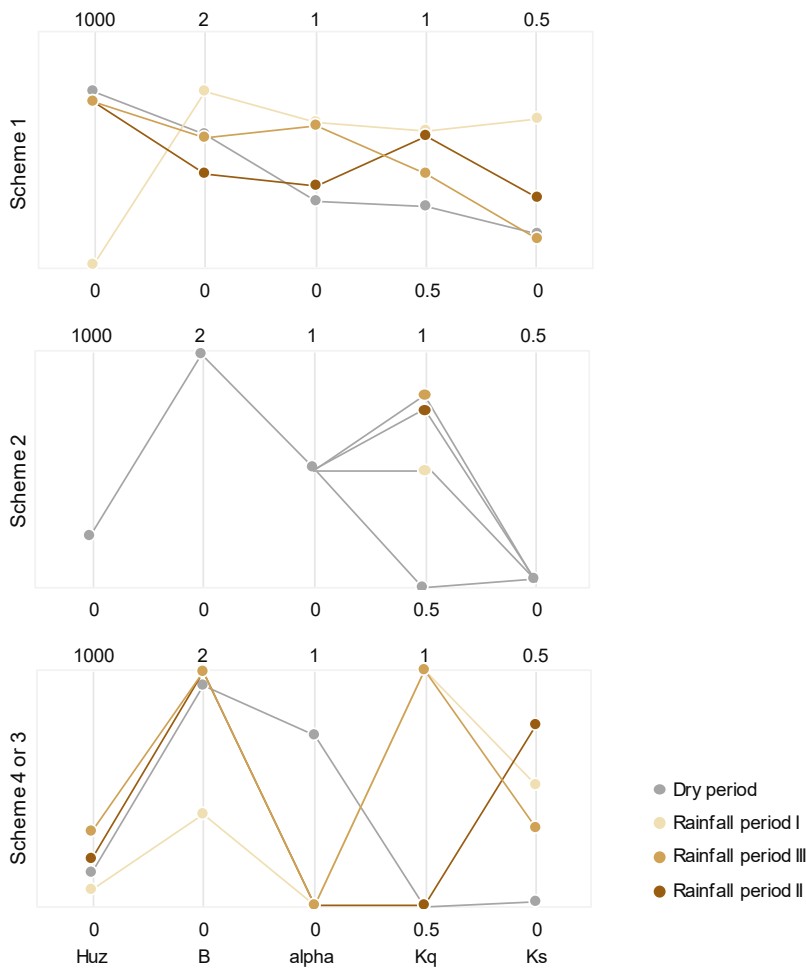

**Figure 5: The dynamized parameter sets optimized by four sub-period calibration schemes in the Hanzhong basin.**



### 4.1.3 State variables and fluxes

As shown in Figure 6 and 7, in Scheme 1, its fluxes and state variables exhibit unreasonable variable values. In Scheme 2, its variables are similar to those in Scheme 0, which implies the running of the model both Scheme 0 and Scheme 2 are similar. This also demonstrated the aforementioned results in section 4.1.1 and 4.1.2, that is, the model performance is only slightly better than that of Scheme 0. In Scheme 3, invalid values are found at the junction of the fluxes and state variables, where the parameter set is abrupt. In Scheme 4, (1) the $Q_s$, $XH_{UZ}$ and $XC_{UZ}$ are lower in the dry period than those in Scheme 0, which is reasonable because the runoff is usually overestimated in the dry period (Pool et al., 2017;Wang et al., 2017a;Tongal and Booij, 2018;Xiong et al., 2018). The values of the $Q_s$, $XH_{UZ}$ and $XC_{UZ}$ are also higher in the rainfall period II than those in Scheme 0, which is logical because the peaks in the rainfall period II are usually underestimated (Guo et al., 2018;Höge et al., 2018;Pande and Moayeri, 2018;Wang et al., 2018). (2) The changes of the state variable $X_s$ and the flux $Q_s$ in Scheme 4 are similar as the aforementioned $Q_s$, $XH_{UZ}$ and $XC_{UZ}$. The reason is that most of the excess streamflow is diverted to the slow-flow tank, hence the state variable $X_s$ and the flux $Q_s$ play more prominent roles in simulating runoff than the quick flow mode in the rainfall period II (wettest period). As a result, the fluxes and state variables in the slow-flow tank mode present higher identifiability in model performance. (3) A comparison of the observations and simulations of the runoff in Scheme 0 and Scheme 4 indicates that both peak flow and flood volume in the rainfall period II are more accurately matched. (4) Moreover, Scheme 4 also contributes to the superior performance in the two normal periods (rainfall period I and rainfall period III). That is, the state variables $XH_{UZ}$ and $XC_{UZ}$ (soil moisture state height and contents) are lower in the rainfall period I and higher in the rainfall period III compared to Scheme 0. This is reasonable because the antecedent soil moisture content in the rainfall period III is higher than in rainfall period I (Lan et al., 2018).

The above results demonstrate again that the model responses to dynamic catchment characteristics are enhanced in Scheme 4. Furthermore, it is interesting to find that the state variables and fluxes are able to describe the dynamic catchment behaviour in a robust manner than the optimal parameters. The results will be discussed in section 4.2.

### 4.2 Possible reasons for poor model performance

The evaluation results of the five schemes based on the multi-metric evaluation, the dynamized parameter values, and the state variables and fluxes are further discussed to explore the possible reasons for poor model performance, as follows:

1. **Time-invariant parameter:** In Scheme 0, the time-invariant parameter set averages the hydrological responses with the simplified model response modes. As a result, Scheme 0 resulted in a poorer simulation accuracy or weaker transferability of the optimized parameters in different flow conditions. The results were consistent with Delorit et al. (2017), Fowler et al. (2018) and Xiong et al. (2019).

2. **Dimension disaster of parameters:** In Scheme 1, it has a sound logic by continuously running the model with the dynamic parameter set like the real system. However, all fluxes and state variables were discontinuous and abnormal, as well as the model presented extremely poor performance. The failure of the modelling run could be likely attributed that



**Figure 6: All fluxes (including $AE$, $OV$, $Q_q$, $Q_s$, and $Q_{sim}$) for five schemes in a reference year in the validation period in Hanzhong basin.**

Note. Variables in different sub-periods are denoted by different colours (same colours as in Figure 2a) to enhance the readability.
The variables of Scheme 0 are denoted by the thin grey lines in each box. The observed streamflow time series data are denoted as thin red lines. All fluxes and state variables in the whole calibration and validation periods are presented in Figure S3, S4, S5, S6, S7, and S8 of the supporting information.





**Figure 7: All state variables (including $XH_{UZ}$ $XC_{UZ}$ $X_{q1}$, $X_{q2}$, $X_{q3}$, and $X_s$) for five schemes in a reference year in the validation period in Hanzhong basin.**

it was difficult to optimize the parameters in high-dimensional parameter space with the interdependency of numerous

5     parameters (Beven and Binley, 1992; Sivakumar, 2004; Bárdossy and Singh, 2008; Laloy and Vrugt, 2012).



3. **"Compensation" among parameter:** In Scheme 2, the dynamization of individual parameters with high identifiability could not effectively respond to dynamic catchment behaviour. The reasons are explained as follows. Bárdossy (2007) demonstrated that the changes in one parameter were compensated for by the changes in other parameters due to their interdependence (Westra et al., 2014;Klotz et al., 2017;Wang et al., 2017b;Wang et al., 2018). Therefore, although the parameter with high identifiability is dynamized over different response modes of models, the other parameters counteracted those changes, resulting in no overall change in the hydrological processes. As a result, the model performance in Scheme 2 did not be improved.

4. **Abrupt shift of parameters:** In Scheme 3, the abrupt shifts of the parameter set at the junction of two sub-periods triggered the anomalous or unreasonable values in the fluxes and state variables time series, which crashed the model running in the validation period. Kim and Han (2017) also emphasize the damage of abrupt shifts of the parameter set on the model running.

In conclusion, Scheme 4 is recommended for sub-period calibration because it captured the temporal variations of the dynamic catchment characteristics and overcome the aforementioned issues for poor model performance. Although Scheme 4 has higher computational cost, this does not represent a much problem with current computing devices.

## 4.3 Evaluation of reliability

The evaluation results of the Scheme 0 and Scheme 4 in individual parameter spaces using ECP-VP in the Hanzhong basin are shown in Figure 8 and as follows:

1. In Scheme 0, (1) the parameter $B$ presents bimodal distribution in whole feasible parameter space, as shown in Figure 8 (a). It represents that the fitness landscape of $B$ is unsteady or fluctuating. The search is difficult to find a global optimum in $B$ parameter space. As a result, the reliability of the optimal $B$ is poor. (2) Although the convergence processes of $H_{UZ}$, $\alpha$, $K_q$, and $K_s$ are quickly controlled in small ranges, their magnified details shown in Figure 8 (b) present the bimodal or multimodal distributions. It is inferred that the convergence processes of these parameters may be affected by prominently local optima. The outcomes of search operations may be arbitrary, which leads the optimizer away from the global optima. As consequence, the reliability of the parameter set optimized by the global optimization algorithm in Scheme 0 is overall poor, and they cannot be used as the indicators effectively characterize the catchment behaviour.

2. In Scheme 4, the four sub-periods are evaluated separately. (1) In the dry period, except for $K_s$, the distributions of other parameters are oscillating in entire feasible parameter space. Indeed, the magnified details of parameter $K_s$ (see Figure 8 (b)) also exhibit multimodal distributions. We find that the parameter $K_s$ not only has the highest identifiability due to slow-flow process dominating the model system in the dry period, but also its best convergence performance. However, due to the weak relationship between precipitation and runoff in the dry period (Moore, 1985;Yadav et al., 2007;De Vos et al., 2010), most modules of the model in the dry period could not effectively characterize the behaviour of the catchment. Consequently, the reliability of the optimal parameter set in the dry period is overall poor. (2) In the rainfall periods I, II,





and III, the parameters $\alpha$ and $K_s$ with unimodal distribution manifest the fastest convergence speed and best convergence levels. The $\alpha$ values in the three rainfall periods are close to the minimum, which means that the slow-flow tank controls the cascade routing component. The $\alpha$ and $K_s$ with high identicality and best convergence performance also indicate that the chosen model is more suitable for the streamflow simulation in the three rainfall periods. The main reason is that the

HYMOD model is well suited for catchments dominated by "saturation excess overland flow" processes. Intense rainfall events contribute to saturation excess overland flow in the rainfall periods (Herman et al., 2013;Sarrazin et al., 2016;Wang et al., 2017a;Wang et al., 2018). Moreover, the results also illustrate that the optimal $\alpha$ and $K_q$ (or $K_s$) in the cascade routing component have higher reliability than the optimal $H_{UZ}$ and $B$ in the soil moisture accounting component. The results from the ECP-VP approach demonstrated the reliability of dynamic parameter set with high convergence

performance in the proposed scheme (Scheme 4). Moreover, the dominant response modes of hydrological models in the different sub-periods are identified. The results are consistent with those of the Hanzhong basin.

The further discussion is that (1) the proposed ECP-VP tool effectively described the convergence behaviour of the models in individual parameter spaces, identifying the reliability of optimized dynamic parameter values. It can also greatly clarify the mechanisms of potentially poor convergence performance. (2) The evaluation of convergence performance in individual

parameter spaces may practically contribute to identifying the operation modes of hydrological models. It provides valuable information on the improvement of hydrological models with different catchment characteristics. (3) Interesting, the results indicate that the convergence performance of all parameters in one sub-period might be superior or inferior to those of other sub-periods. For example, the convergence performance of all parameters was worse in the dry period than those in the three rainfall periods. Indeed, due to the complicated correlations between the parameters in a parameter set, the convergence

performance in an individual parameter may be significantly affected by the other parameters. For this reason, it is not recommended to use the convergence performance of individual parameters but rather the convergence performance of the parameter set. However, the application of this solution requires a significant amount of experiments, validation, analysis, and discussion and these points will be investigated in future studies. (4) Figure 8 illustrates that the convergence behaviour for the same parameter in the different sub-periods is quite different. It further demonstrates that the response systems of

hydrological models to the catchment behaviour on diverse catchment characteristics are significantly different. In addition, the evaluation results of ECP-VP for Scheme 0 and Scheme 4 in the Mumahe basin and Xunhe basin are shown in Figure S9 and Figure S10 in the supporting information. The results are similar to those of the Hanzhong basin.

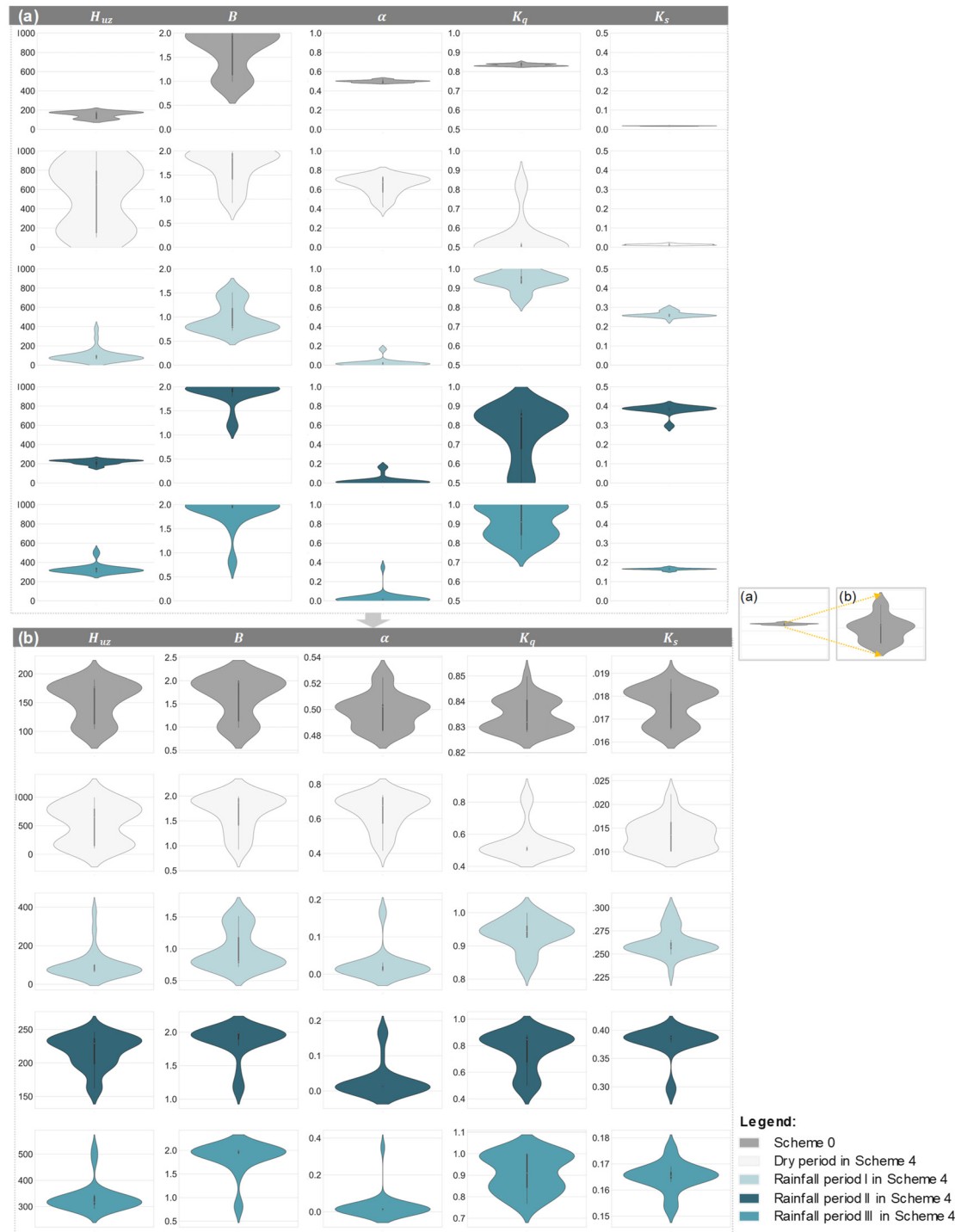

**Figure 8: Convergence performance for Scheme 0 and Scheme 4 in individual parameter spaces in Hanzhong basin using ECP-VP approach.**

Note. (a) The convergence processes in whole feasible parameter spaces; (b) the magnified convergence processes of the parameters.





## 5 Conclusions

We proposed a reliable calibration scheme for dynamics of hydrological model parameters, regarding objective functions, parameters, the fluxes and state variables of hydrological models, and explored the possible reasons for poor model performance. Furthermore, a simple and powerful approach to visualize and evaluate the convergence processes using violin
plots (ECP-VP) was developed to assess the reliability of the dynamic parameter set. The following conclusions can be drawn:

1.   Five designed schemes were systematically evaluated with respect to multi-meristic evaluation, dynamized parameter values, state variables, and fluxes. The best scheme was recommended for the dynamization of the parameter set. According to the evaluation results, the possible reasons for the poor model performance are explored as follows: (1) time-constant parameters, (2) dimension disaster of parameters (3) "compensation" among parameter, and (4) abrupt shift
of parameters. Interestingly, the results also proved that changes in the state variables and fluxes time series provided a more robust description of the dynamic catchment characteristics than the optimal parameters.

2.   The proposed calibration (1) compensates for the deficiencies of model structure; (2) it provided the high forecast accuracy for different flow phases; (3) it exhibited good transferability of the model parameters in the calibration-validation period; (4) it enhanced the identification of the dominant parameters and their responses to catchment processes;
(5) it accurately characterized the dynamic behaviour of catchments.

3.   The proposed ECP-VP framework regarding possible properties' sketches of fitness landscapes could greatly investigate the convergence behaviour in individual parameter spaces, and evaluate the reliability of the optimal parameters. The evaluation results for dynamized parameters using the ECP-VP also provides valuable information on recognizing the dominant response modes of hydrological models in different sub-periods or catchments with the distinguishing
catchment characteristics. In addition, the convergence performance of a certain parameter can be affected by the convergence performance of other parameters due to the intricate correlations among parameters.

### Acknowledgments

This study is financially supported by the Excellent Young Scientist Foundation of NSFC (51822908), the National Natural Science Foundation of China (No. 51779279), the National Key R&D Program of China (2017YFC0405900), Open research
foundation of Dynamics and the associated process control key laboratory in the pearl river estuary of ministry of water resources (2017KJ12), Baiqianwan project's young talents plan of special support program in Guangdong Province (42150001), and the Research Council of Norway (FRINATEK Project 274310). Digital Elevation Model (DEM) of the study area is derived from the Advanced Spaceborne Thermal Emission and Reflection Radiometer (ASTER) global digital elevation model (GDEM) with a cell size of 30 × 30 m which are obtained from https://asterweb.jpl.nasa.gov/. The climatic datasets consist of
daily rainfall datasets and pan evaporation datasets provided by the China Climatic Data Sharing Service System which are obtained from https://data.cma.cn/en. Daily streamflow used to support this paper can be made available for interested readers by the corresponding author at linkr@mail.sysu.edu.cn.



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
