# Peer review of "Dynamics of hydrological model parameters: calibration and reliability"

_Hydrology and Earth System Sciences, 2019_

## Referee Comment (RC1) · Anonymous Referee #1 · 5 Aug 2019

**Reviews of HESS-2019-301**

Title: Dynamics of hydrological model parameters: calibration and reliability

The objective of this paper is two-hold: 1) to develop and test the strategies of sub-period model calibration to generate temporarily varying optimized parameter sets and 2) develop the method to assess reliability of the optimized parameter set by evaluating parameter convergence behaviors. The paper presents the calibration results for one Chinese basin (two more in supplemental material) with focus on those two goals.

My first impression was the paper lacks focus due to two different objectives (distracted by each other), so I would lean to suggest putting more focus on the first goal and then reduce tone on the issue on parameter convergence evaluation. However, I found the results presented in this paper are interesting and reasonable overall. I have several specific comments below. In addition, the authors should work on textural improvements including fixing grammatical errors, punctuations (excessive use of parenthesis), vocabulary, and most importantly, more conciseness throughout the paper. I think the manuscript requires major revision before publication.

**Specific comments:**

- I am not sure about a list of the past studies on sub-period calibration (Page 2 Line 23-24). The most relevant paper to this study would be *Merz et al.*, [2011]. Please re-evaluate which references should be relevant to this sub-period calibration topic. Introduction should emphasize this topic than convergency behavior.
- Method for partitioning of the simulation period into sub-periods are not described in this paper but seems to be climatological based, i.e., dry, wet periods, and backbone for this calibration strategy. There is no information on this. Although a great deal of this topic is in another publication by the same author, I would like to see some summary of the paper, including what variables are used for clustering, and very brief clustering methods.
- For scheme 2. I understood that this is the same as scheme 1 except that one selected parameter is optimized per sub-period and the others are optimized for the entire simulation period. It is not clear to me what the motivation for this scheme is. And also wonder which parameter is exposed to sub-period calibration and how it is selected? Please clarify.
- Minor comments on the figures. Figure 2: I am not sure panel c is needed. It does not add anything meaningful to me. Table 1 would be enough. Figure 3. Panel c is specific to SCE and not general and I don't understand well about panel d. Figure 4. I don't think panel b is necessary. Also, RMSE for FDC is normalized by something?
- The methodology of parameter convergence assessment (3.2.2) is very specific to SCE, but not seems to be for the other algorithms. I think the concept works for the other global evolution algorithms, including DDS and even for multi-objective algorithms. My

recommendation is to generalize more technical descriptions on the procedures so that it is more applicable to such other algorithms.

- The most of hydrologic models struggle with dry basin calibration. For US basin, see *Newman et al.*, 2015, 2017. Interestingly Figure 8 shows dry period calibration also struggle converging the optimizing parameter values. I think this is something to discuss and would suggest showing (or mentioning) performance metrics for each 4 period for Scheme 3 and 4. My speculation is much better performance metrics for the wet periods than dry period, and reason why scheme 0 and 1 produce poor performance is due to poor performance during the dry period.
- Merz, R., J. Parajka, and G. Blöschl (2011), Time stability of catchment model parameters: Implications for climate impact analyses, *Water Resour. Res.*, *47*(2), doi:10.1029/2010WR009505.
- Newman, A. J. et al. (2015), Development of a large-sample watershed-scale hydrometeorological data set for the contiguous USA: data set characteristics and assessment of regional variability in hydrologic model performance, *Hydrol. Earth Syst. Sci.*, *19*(1), 209–223, doi:10.5194/hess-19-209-2015.
- Newman, A. J., N. Mizukami, M. P. Clark, A. W. Wood, B. Nijssen, and G. Nearing (2017), Benchmarking of a Physically Based Hydrologic Model, *J. Hydrometeorol.*, *18*, 2215–2225, doi:10.1175/JHM-D-16-0284.1.

---

## Referee Comment (RC2) · Anonymous Referee #2 · 13 Aug 2019

Review of paper HESS-2019-301

**Title:** Dynamics of hydrological model parameters: calibration and reliability

This paper presents a methodology for the estimation of parameter sets of hydrological models that vary in time. The main argument is that by considering the model parameters dynamic throughout different simulation periods, hydrological models can better represent the dynamic behavior observed in real catchments. The paper also presents a methodology to assess the reliability and performance of the optimization.

The authors apply the proposed method to three basins located in China, for which different sub-periods with different hydrological characteristics were previously defined (research done by the same authors).

In the following, I provide general remarks on this paper, and afterwards a list of more specific comments.

**General comments**

I find the work presented in this paper of relevance and major interest for the scientific community. Although the benefit of considering time-varying parameters in hydrological modeling has been highlighted in many publications, considering dynamic parameter sets during model calibration has not yet been given great attention. The topic discussed within this work fits the scope of HESS. However, the authors need to do a thorough proofread of the paper. Unfortunately, the grammatical errors, confusing sentences, redundant vocabulary and an erratic writing style, hinder the message that the authors want to convey, and in some cases render some statements ambiguous or even mistaken.

I conclude that this work cannot be considered for publication as it is. I recommend the authors to further work with the text and structure of the manuscript, and encourage to undergo a resubmission process. I would be more than willing to continue the review process once a new improved version of the manuscript is available.

**Specific comments**

*For Section 2. Background*

- The description of the previous research is poorly presented. I suggest merging *section 1* of the supplement with *Section 2 Background*, and include relevant information concerning the clustering method and the main results that led to the definition of the sub-periods in the three sub-basins.

In agreement with referee 1, I consider that the second objective defined by the authors shadows the first one. The suggested approach to assess the convergence performance of the optimization should be considered as a tool chosen by the authors, and not as one of the main objectives of the work. Still, the advantages of such an assessment tool over others should be emphasized.

*For Section 3.1.1 Sub-period calibration schemes.*

- Explanation of the sub-period calibration schemes is confusing, vague wording.
- I suggest adding at the beginning of the subsection a synthetized and general description of figure 2, guiding the reader through such a complex figure. I got the impression that the three arrows in figure 2b are related to the objective function, parameters, and state variables or fluxes

compartments of subfigure 2a. If that is the case, the alignment between figure 2a and b should be fixed. Ultimately, not sure whether subfigures 2b and 2c are really necessary.

- For scheme 2, how do the authors define which parameter is to be dynamic and which parameters are fixed throughout the calibration?

*For Section 3.2.2 A tool for reliability evaluation.*

- If the method to assess parameter convergence is designed specifically for SCE, I suggest to elaborate in the description of the theory behind SCE, otherwise is hard to understand how does the assessment tool really functions.
- Following the previous comment, I consider subfigure Figure 3c not necessary if SCE is not really explained in the text.

---

## Author Comment (AC1) · 10 Sep 2019

Dear Anonymous Referee #1,

Re: Manuscript #HESS-2019-301 entitled "Dynamics of hydrological model parameters: calibration and reliability".

Many thanks for your positive evaluation, encouragement for the results and scientific significance in this study. We greatly appreciate the Referee's comments, especially in the focus of the first goal and textural improvements. All suggestions are helpful to improve this manuscript.

We have carefully studied, considered and responded to all comments point-by-point as follows. For clarity, all comments are given in black and responses are given in

the blue text. All the comments and suggestions have been replied below and will be addressed in the revision.

Yours sincerely,

Kairong Lin (Ph.D.) Professor in hydrology E-mail: linkr@mail.sysu.edu.cn

Please also note the supplement to this comment:
https://www.hydrol-earth-syst-sci-discuss.net/hess-2019-301/hess-2019-301-AC1-supplement.pdf

**Supplement:**

**Replies to Referee #1**

Dear Anonymous Referee #1,

Re: Manuscript #HESS-2019-301 entitled "Dynamics of hydrological model parameters: calibration and reliability".

Many thanks for your positive evaluation, encouragement for the results and scientific significance in this study. We greatly appreciate the Referee's comments, especially in the focus of the first goal and textural improvements. All suggestions are helpful to improve this manuscript.

We have carefully studied, considered and responded to all comments point-by-point as follows. For clarity, all comments are given in black and responses are given in the blue text. All the comments and suggestions have been replied below and will be addressed in the revision.

Yours sincerely,

Kairong Lin (Ph.D.)
Professor in hydrology
E-mail: linkr@mail.sysu.edu.cn

**Title: Dynamics of hydrological model parameters: calibration and reliability**

The objective of this paper is two-hold: 1) to develop and test the strategies of sub-period model calibration to generate temporarily varying optimized parameter sets and 2) develop the method to assess reliability of the optimized parameter set by evaluating parameter convergence behaviors. The paper presents the calibration results for one Chinese basin (two more in supplemental material) with focus on those two goals.

**General comments:**

My first impression was the paper lacks focus due to two different objectives (distracted by each other), so I would lean to suggest putting more focus on the first goal and then reduce tone on the issue on parameter convergence evaluation. However, I found the results presented in this paper are interesting and reasonable overall. I have several specific comments below. In addition, the authors should work on textural improvements including fixing grammatical errors, punctuations (excessive use of parenthesis), vocabulary, and most importantly, more conciseness throughout the paper. I think the manuscript requires major revision before publication.

**Reply:** We appreciate that the Referee is in favor of the content of this research. We agree and follow the suggestion of reviewer, more focus will be paid on enhancing the first objective in the revised version. Meanwhile, the parameter convergence evaluation (currently the second objective) will be regarded as a tool, and not as one of the main goals in this work. The detailed description in this topic will be moved to the supplementary materials. We will do a thorough revision of this paper to improve the presentation quality. Besides, the English will be corrected by a professional before submission of the revision.

**Specific comments:**

- I am not sure about a list of the past studies on sub-period calibration (Page 2 Line 23-24). The most relevant paper to this study would be *Merz et al*., [2011]. Please reevaluate which references should be relevant to this sub-period calibration topic. Introduction should emphasize this topic than convergency behavior.

**Reply:** We thank the reviewer for the suggestion and comment. We have studied the paper by *Merz et al*. [2011] and found it is much relevant to our study. Hence, we will discuss this paper in the *Introduction* section of revised manuscript. Moreover, all references in the sub-period calibration topic will be reevaluated in the revised manuscript.

We agree with the Referee's comment that *Introduction* should emphasize the sub-period calibration schemes than the assessment of convergency behavior. The introduction in the sub-period calibration section will be supplemented and improved in the revised manuscript. Meanwhile, the content concerning the parameter convergence evaluation will be shortened in *Introduction* section and details will be moved to the supplementary materials.

- Method for partitioning of the simulation period into sub-periods are not described in this paper but seems to be climatological based, i.e., dry, wet periods, and backbone for this calibration strategy. There is no information on this. Although a great deal of this topic is in another publication by the same author, I would like to see some summary of the paper, including what variables are used for clustering, and very brief clustering methods.

**Reply:** We agree with the Referee's comment. The method for clustering the simulation period into sub-period in another publication by the same authors will be concisely summarized in the revised manuscript. In addition, the specific explanation will be also presented in the supplementary materials.

- For scheme 2. I understood that this is the same as scheme 1 except that one selected parameter is optimized per sub-period and the others are optimized for the entire simulation period. It is not clear to me what the motivation for this scheme is. And also wonder which parameter is exposed to sub-period calibration and how it is selected? Please clarify.

**Reply:** Thanks for the Referee's comment. For scheme 2, the parameters which are sensitive to dynamic catchment characteristics were usually chosen to calibrate the models. However, due to the complex correlations among the parameters, the individual parameters may not represent their defined physical characteristics. Hence, the most sensitive parameters were usually identified and optimized per sub-period, and the others are optimized for the entire simulation period (Merz et al., 2011; Me et al., 2015; Pfannerstill et al., 2015; Zhang et al., 2015; Deng et al., 2016; Guse et al., 2016; Ouyang et al., 2016; Deng et al., 2018; Xiong et al., 2019). In this regard, the most sensitive parameter $K_q$ identified by the HYMOD application carried in the study areas was selected to sub-period calibration in this work. All related explanation will be clarified in the revised manuscript.

Moreover, considering the possible interference in calibration artifacts (Merz et al., 2011), all parameters in HYMOD will be exposed to sub-period calibration, respectively. The relevant discussion will be supplemented into the revised manuscript.

References:

Deng, C., Liu, P., Guo, S. L., Li, Z. J., and Wang, D. B.: Identification of hydrological model parameter variation using ensemble Kalman filter, Hydrol Earth Syst Sc, 20, 4949-4961, https://doi.org/10.5194/hess-20-4949-2016, 2016.

Deng, C., Liu, P., Wang, D. B., and Wang, W. G.: Temporal variation and scaling of parameters for a monthly hydrologic model, J Hydrol, 558, 290-300, https://doi.org/10.1016/j.jhydrol.2018.01.049, 2018.

Guse, B., Pfannerstill, M., Strauch, M., Reusser, D. E., Lüdtke, S., Volk, M., Gupta, H., and Fohrer, N.: On characterizing the temporal dominance patterns of model parameters and processes, Hydrol Process, 30, 2255-2270, https://doi.org/10.1002/hyp.10764, 2016.

Me, W., Abell, J. M., and Hamilton, D. P.: Effects of hydrologic conditions on SWAT model performance and parameter sensitivity for a small, mixed land use catchment in New Zealand, Hydrol Earth Syst Sc, 19, 4127-4147, https://doi.org/10.5194/hess-19-4127-2015, 2015.

Merz, R., Parajka, J., and Blöschl, G.: Time stability of catchment model parameters: Implications for climate impact analyses, 47, 10.1029/2010wr009505, 2011.

Ouyang, Y., Xu, D., Leininger, T. D., and Zhang, N.: A system dynamic model to estimate hydrological processes and water use in a eucalypt plantation, Ecological Engineering, 86, 290-299, 10.1016/j.ecoleng.2015.11.008, 2016.

Pfannerstill, M., Guse, B., Reusser, D., and Fohrer, N.: Process verification of a hydrological model using a temporal parameter sensitivity analysis, Hydrol Earth Syst Sc, 19, 4365-4376, https://doi.org/10.5194/hess-19-4365-2015, 2015.

Xiong, M., Liu, P., Cheng, L., Deng, C., Gui, Z., Zhang, X., and Liu, Y.: Identifying time-varying hydrological

model parameters to improve simulation efficiency by the ensemble Kalman filter: A joint assimilation of streamflow and actual evapotranspiration, J Hydrol, 568, 758-768, https://doi.org/10.1016/j.jhydrol.2018.11.038, 2019.

Zhang, D., Chen, X., Yao, H., and Lin, B.: Improved calibration scheme of SWAT by separating wet and dry seasons, Ecol Model, 301, 54-61, https://doi.org/10.1016/j.ecolmodel.2015.01.018, 2015.

- Minor comments on the figures. Figure 2: I am not sure panel c is needed. It does not add anything meaningful to me. Table 1 would be enough. Figure 3. Panel c is specific to SCE and not general and I don't understand well about panel d. Figure 4. I don't think panel b is necessary. Also, RMSE for FDC is normalized by something?

**Reply:** We agree with the Referee's suggestion. The panel c in Figure 2 and panel b in Figure 4 will be removed. The SCE-UA algorithm is a subset of global evolution algorithms (see Figure S1) (Duan et al., 1993; Hanne, 2000; Michalewicz and Schoenauer, 1996; Omran and Mahdavi, 2008; Storn and Price, 1997; Yiu-Wing and Yuping, 2001). The method to assess parameter convergence is designed generally for global evolution algorithms. The panel c in Figure 3 will be revised and the general applicability of the methodology to assess parameter convergence will be elaborated in the revised manuscript.

[Figure]

**Figure S1:** The basic cycle of global evolution algorithms.
*Note.* Initial population: Create an initial population of random individuals; Evaluation: Compute the objective values of the solution candidates; Fitness assignment: Use the objective values to determine fitness values; Selection: Select the fittest individuals for reproduction; Reproduction: Create new individuals from the mating pool by crossover and mutation.

The panel d in Figure 4 illustrated that the convergence process evolves toward minimizing the objective function values. The convergence speed can be assessed by the number of iterations. The ambiguous explanation will be modified in the revised manuscript.

A multi-metric framework is conducted to assess the prediction accuracy of various flow conditions. The metrics incorporate the NSE, the NSE of the logarithmic streamflow (LNSE), and a five-segment flow duration curve (5FDC) with the RMSE. Its elaboration has been presented in the supplementary materials. Furthermore, the multi-metric framework will be summarized in the revised manuscript.

References:

Duan, Q. Y., Gupta, V. K., Sorooshian, S. J. J. o. O. T., and Applications: Shuffled complex evolution approach for effective and efficient global minimization, 76, 501-521, 10.1007/bf00939380, 1993.

Hanne, T. J. J. o. H.: Global Multiobjective Optimization Using Evolutionary Algorithms, 6, 347-360, 10.1023/a:1009630531634, 2000.

Michalewicz, Z., and Schoenauer, M.: Evolutionary Algorithms for Constrained Parameter Optimization Problems, 4, 1-32, 10.1162/evco.1996.4.1.1, 1996.

Omran, M. G. H., and Mahdavi, M.: Global-best harmony search, Applied Mathematics and Computation, 198, 643-656, https://doi.org/10.1016/j.amc.2007.09.004, 2008.

Storn, R., and Price, K. J. J. o. G. O.: Differential Evolution – A Simple and Efficient Heuristic for global Optimization over Continuous Spaces, 11, 341-359, 10.1023/a:1008202821328, 1997.

Yiu-Wing, L., and Yuping, W.: An orthogonal genetic algorithm with quantization for global numerical optimization, Ieee T Evolut Comput, 5, 41-53, 10.1109/4235.910464, 2001.

- The methodology of parameter convergence assessment (3.2.2) is very specific to SCE, but not seems to be for the other algorithms. I think the concept works for the other global evolution algorithms, including DDS and even for multi-objective algorithms. My recommendation is to generalize more technical descriptions on the procedures so that it is more applicable to such other algorithms.

**Reply:** We really appreciate your advice. The SCE-UA algorithm will be replaced by the basic concepts of generally global evolution algorithms, as shown in Figure S1. The more technical descriptions will be added to the revised manuscript.

- The most of hydrologic models struggle with dry basin calibration. For US basin, see Newman et al., 2015, 2017. Interestingly Figure 8 shows dry period calibration also struggle converging the optimizing parameter values. I think this is something to discuss and would suggest showing (or mentioning) performance metrics for each 4 period for Scheme 3 and 4. My speculation is much better performance metrics for the wet periods than dry period, and reason why scheme 0 and 1 produce poor performance is due to poor performance during the dry period.

Merz, R., J. Parajka, and G. Blöschl (2011), Time stability of catchment model parameters: Implications for climate impact analyses, *Water Resour. Res., 47*(2), doi:10.1029/2010WR009505.

Newman, A. J. et al. (2015), Development of a large-sample watershed-scale hydrometeorological data set for the contiguous USA: data set characteristics and assessment of regional variability in hydrologic model performance, *Hydrol. Earth Syst. Sci., 19*(1), 209–223, doi:10.5194/hess-19-209-2015.

Newman, A. J., N. Mizukami, M. P. Clark, A. W. Wood, B. Nijssen, and G. Nearing (2017), Benchmarking of a Physically Based Hydrologic Model, *J. Hydrometeorol., 18*, 2215–2225, doi:10.1175/JHM-D-16-0284.1.

**Reply:** Thanks for your valuable suggestions for the relationship between dry period calibration and parameter convergence performance. The performance metrics for 4 periods for Schemes 3 and 4 will be added, and the reasons why the scheme 0 and 1 produce poor performance will be discussed in the revision.

---

## Author Comment (AC2) · 10 Sep 2019

Dear Anonymous Referee #2,

Re: Manuscript #HESS-2019-301 entitled "Dynamics of hydrological model parameters: calibration and reliability".

We sincerely appreciate the referee is favor of the content of this research and the positive evaluation for its scientific significance. The Referee's constructive suggestions for background, sub-period calibration schemes, and a tool for reliability evaluation are helpful to improve this manuscript. Most importantly, we would make efforts to improve the text and structure of this manuscript. We are very grateful for your great patience on a new improved version of the manuscript.

[Figure]

We have carefully studied, considered and responded to all comments point-by-point as follows. For clarity, all comments are given in black and responses are given in the blue text. All the comments and suggestions have been replied below and will be addressed in the revision.

Yours sincerely,

Kairong Lin (Ph.D.) Professor in hydrology E-mail: linkr@mail.sysu.edu.cn

Please also note the supplement to this comment:
https://www.hydrol-earth-syst-sci-discuss.net/hess-2019-301/hess-2019-301-AC2-supplement.pdf

―――――――――――――――

**Supplement:**

**Replies to Referee #2**

Dear Anonymous Referee #2,

Re: Manuscript #HESS-2019-301 entitled "Dynamics of hydrological model parameters: calibration and reliability".

We sincerely appreciate the referee is favor of the content of this research and the positive evaluation for its scientific significance. The Referee's constructive suggestions for *background*, *sub-period calibration schemes*, and *a tool for reliability evaluation* are helpful to improve this manuscript. Most importantly, we would make efforts to improve the text and structure of this manuscript. We are very grateful for your great patience on a new improved version of the manuscript.

We have carefully studied, considered and responded to all comments point-by-point as follows. For clarity, all comments are given in black and responses are given in the blue text. All the comments and suggestions have been replied below and will be addressed in the revision.

Yours sincerely,

Kairong Lin (Ph.D.)
Professor in hydrology
E-mail: linkr@mail.sysu.edu.cn

**General comments:**

I find the work presented in this paper of relevance and major interest for the scientific community. Although the benefit of considering time-varying parameters in hydrological modeling has been highlighted in many publications, considering dynamic parameter sets during model calibration has not yet been given great attention. The topic discussed within this work fits the scope of HESS. However, the authors need to do a thorough proofread of the paper. Unfortunately, the grammatical errors, confusing sentences, redundant vocabulary and an erratic writing style, hinder the message that the authors want to convey, and in some cases render some statements ambiguous or even mistaken.

I conclude that this work cannot be considered for publication as it is. I recommend the authors to further work with the text and structure of the manuscript and encourage to undergo a resubmission process. I would be more than willing to continue the review process once a new improved version of the manuscript is available.

**Reply:** We greatly appreciate the positive evaluation for scientific significance of this study. With your constructive suggestions, the revised version will be greatly improved, especially in the presentation quality. Besides, the English will be corrected by a professional before submission of the revision.

**Specific comments:**

*For Section 2. Background*

- The description of the previous research is poorly presented. I suggest merging section 1 of the supplement with Section 2 Background, and include relevant information concerning the clustering method and the main results that led to the definition of the sub-periods in the three sub-basins.

In agreement with referee 1, I consider that the second objective defined by the authors shadows the first one. The suggested approach to assess the convergence performance of the optimization should be considered as a tool chosen by the authors, and not as one of the main objectives of the work. Still, the advantages of such an assessment tool over others should be emphasized.

**Reply:** We agree with the Referee's comment. Section 1 and section 2 will be merged in the revised manuscript. The relevant clustering method, the definition of the sub-period, and the main results in the study areas will be supplemented in the revised manuscript. Meanwhile, the parameter convergence evaluation (currently the second objective) will be regarded as a tool, and not as one of the main goals in this work. The detailed description in this topic will be moved to the supplementary materials.

*For Section 3.1.1 Sub-period calibration schemes.*

- Explanation of the sub-period calibration schemes is confusing, vague wording.
- I suggest adding at the beginning of the subsection a synthetized and general description of figure 2, guiding the reader through such a complex figure. I got the impression that the three arrows in figure 2b are related to the objective function, parameters, and state variables or fluxes compartments of subfigure 2a. If that is the case, the alignment between figure 2a and b should be fixed. Ultimately, not sure whether subfigures 2b and 2c are really necessary.

**Reply:** Thank you for the Referee's constructive advice. A synthesized and general description of the four sub-period calibration schemes will be added in the caption of Figure 2. We agree with Referee's comment that the subfigures 2b and 2c will be removed in the revised manuscript.

- For scheme 2, how do the authors define which parameter is to be dynamic and which parameters are fixed throughout the calibration?.

**Reply:** Thanks for the Referee's comment. For scheme 2, the parameters which are sensitive to dynamic catchment characteristics were usually chosen to calibrate the models. However, due to the complex correlations among the parameters, the individual parameter may not represent its defined physical characteristics. Hence, the most sensitive parameters were usually identified and optimized per sub-period, and the others are optimized for the entire simulation period (Merz et al., 2011; Me et al., 2015; Pfannerstill et al., 2015; Zhang et al., 2015; Deng et al., 2016; Guse et al., 2016; Ouyang et al., 2016; Deng et al., 2018; Xiong et al., 2019). Accordingly, the most sensitive parameter $K_q$ identified by the HYMOD application carried in the study areas was selected to sub-period calibration in this work. All related explanation will be clarified in the revised manuscript.

Moreover, considering the possible interference in calibration artifacts (Merz et al., 2011), all parameters in HYMOD will be exposed to sub-period calibration, respectively. The relevant discussion will be supplemented into the revised manuscript.

References:

Deng, C., Liu, P., Guo, S. L., Li, Z. J., and Wang, D. B.: Identification of hydrological model parameter variation using ensemble Kalman filter, Hydrol Earth Syst Sc, 20, 4949-4961, https://doi.org/10.5194/hess-20-4949-2016, 2016.

Deng, C., Liu, P., Wang, D. B., and Wang, W. G.: Temporal variation and scaling of parameters for a monthly hydrologic model, J Hydrol, 558, 290-300, https://doi.org/10.1016/j.jhydrol.2018.01.049, 2018.

Guse, B., Pfannerstill, M., Strauch, M., Reusser, D. E., Lüdtke, S., Volk, M., Gupta, H., and Fohrer, N.: On characterizing the temporal dominance patterns of model parameters and processes, Hydrol Process, 30, 2255-2270, https://doi.org/10.1002/hyp.10764, 2016.

Me, W., Abell, J. M., and Hamilton, D. P.: Effects of hydrologic conditions on SWAT model performance and parameter sensitivity for a small, mixed land use catchment in New Zealand, Hydrol Earth Syst Sc, 19, 4127-4147, https://doi.org/10.5194/hess-19-4127-2015, 2015.

Merz, R., Parajka, J., and Blöschl, G.: Time stability of catchment model parameters: Implications for climate impact analyses, 47, 10.1029/2010wr009505, 2011.

Ouyang, Y., Xu, D., Leininger, T. D., and Zhang, N.: A system dynamic model to estimate hydrological processes and water use in a eucalypt plantation, Ecological Engineering, 86, 290-299, 10.1016/j.ecoleng.2015.11.008, 2016.

Pfannerstill, M., Guse, B., Reusser, D., and Fohrer, N.: Process verification of a hydrological model using a temporal parameter sensitivity analysis, Hydrol Earth Syst Sc, 19, 4365-4376, https://doi.org/10.5194/hess-19-4365-2015, 2015.

Xiong, M., Liu, P., Cheng, L., Deng, C., Gui, Z., Zhang, X., and Liu, Y.: Identifying time-varying hydrological model parameters to improve simulation efficiency by the ensemble Kalman filter: A joint assimilation of streamflow and actual evapotranspiration, J Hydrol, 568, 758-768, https://doi.org/10.1016/j.jhydrol.2018.11.038, 2019.

Zhang, D., Chen, X., Yao, H., and Lin, B.: Improved calibration scheme of SWAT by separating wet and dry seasons, Ecol Model, 301, 54-61, https://doi.org/10.1016/j.ecolmodel.2015.01.018, 2015.

*For Section 3.2.2 A tool for reliability evaluation.*
- If the method to assess parameter convergence is designed specifically for SCE, I suggest to elaborate in the description of the theory behind SCE, otherwise is hard to understand how does the assessment tool really functions.
- Following the previous comment, I consider subfigure Figure 3c not necessary if SCE is not really explained in the text.

**Reply:** We really appreciate your advice. The SCE-UA algorithm is a subset of global evolution algorithms (see Figure S1) (Duan et al., 1993; Hanne, 2000; Michalewicz and Schoenauer, 1996; Omran and Mahdavi, 2008; Storn and Price, 1997; Yiu-Wing and Yuping, 2001). The method to assess parameter convergence is designed generally for global evolution algorithms. The SCE-UA algorithm will be replaced by the basic concepts of generally global evolution algorithms, as shown in Figure S1. The more technical descriptions will be added to the revised manuscript. Moreover, the specific theories of SCE-UA algorithm will also be added in the supplementary materials.

[Figure]

**Figure S1:** The basic cycle of global evolution algorithms.
*Note.* Initial population: Create an initial population of random individuals; Evaluation: Compute the objective values of the solution candidates; Fitness assignment: Use the objective values to determine fitness values; Selection: Select the fittest individuals for reproduction; Reproduction: Create new individuals from the mating pool by crossover and mutation.

References:

Duan, Q. Y., Gupta, V. K., Sorooshian, S. J. J. o. O. T., and Applications: Shuffled complex evolution approach for effective and efficient global minimization, 76, 501-521, 10.1007/bf00939380, 1993.

Hanne, T. J. J. o. H.: Global Multiobjective Optimization Using Evolutionary Algorithms, 6, 347-360, 10.1023/a:1009630531634, 2000.

Michalewicz, Z., and Schoenauer, M.: Evolutionary Algorithms for Constrained Parameter Optimization Problems, 4, 1-32, 10.1162/evco.1996.4.1.1, 1996.

Omran, M. G. H., and Mahdavi, M.: Global-best harmony search, Applied Mathematics and Computation, 198, 643-656, https://doi.org/10.1016/j.amc.2007.09.004, 2008.

Storn, R., and Price, K. J. J. o. G. O.: Differential Evolution – A Simple and Efficient Heuristic for global Optimization over Continuous Spaces, 11, 341-359, 10.1023/a:1008202821328, 1997.

Yiu-Wing, L., and Yuping, W.: An orthogonal genetic algorithm with quantization for global numerical optimization, Ieee T Evolut Comput, 5, 41-53, 10.1109/4235.910464, 2001.